# Exploring the Promise and Limits of Real-Time Recurrent Learning

## Abstract

Real-time recurrent learning (RTRL) for sequence-processing recurrent neural net-
works (RNNs) offers certain conceptual advantages over backpropagation through
time (BPTT). RTRL requires neither caching past activations nor truncating con-
text, and enables online learning. However, RTRL's time and space complexity
makes it impractical. To overcome this problem, most recent work on RTRL fo-
cuses on approximation theories, while experiments are often limited to diagnostic
settings. Here we explore the practical promise of RTRL in more realistic settings.
We study actor-critic methods that combine RTRL and policy gradients, and test
them in several subsets of DMLab-30, ProcGen, and Atari-2600 environments. On
DMLab memory tasks, our system is competitive with or outperforms well-known
IMPALA and R2D2 baselines trained on 10 B frames, while using fewer than 1.2 B
environmental frames. To scale to such challenging tasks, we focus on certain well-
known neural architectures with element-wise recurrence, allowing for tractable
RTRL without approximation. We also discuss rarely addressed limitations of
RTRL in real-world applications, such as its complexity in the multi-layer case.[1]

## 1   Introduction

There are two classic learning algorithms to compute exact gradients for sequence-processing recur-
rent neural networks (RNNs): real-time recurrent learning (RTRL; [1, 2, 3, 4]) and backpropagation
through time (BPTT; [5, 6]) (reviewed in Sec. 2). In practice, BPTT is the only one commonly used
today, simply because BPTT is tractable while RTRL is not. In fact, the time and space complexities
of RTRL for a fully recurrent NN are quadratic and cubic in the number of hidden units, respectively,
which are prohibitive for any RNNs of practical sizes in real applications. Despite such an obvious
complexity bottleneck, RTRL has certain attractive conceptual advantages over BPTT. BPTT requires
to cache activations for each new element of the sequence processed by the model, for later gradient
computation. As the amount of these past activations to be stored grows linearly with the sequence
length, practitioners (constrained by the actual memory limit of their hardware) use the so-called *trun-
cated* BPTT (TBPTT; [7]) where they specify the maximum number of time steps for this storage,
giving up gradient components—and therefore credit assignments—that go beyond this time span. In
contrast, RTRL does not require storing past activations, and enables computation of untruncated
gradients for sequences of any arbitrary length. In addition, RTRL is an online learning algorithm
(more efficient than BPTT to process long sequences in the online scenario) that allows for updating
weights immediately after consuming every new input (assuming that the external error feedback to
the model output is also available for each input). These attractive advantages of RTRL still actively
motivate researchers to work towards practical RTRL (e.g., [8, 9, 10, 11, 12]).

---

[1]Upon acceptance, we will add a GitHub link to our public code here.

The root of RTRL's high complexities is the computation and storage of the so-called *sensitivity matrix* whose entries are derivatives of the hidden activations w.r.t. each trainable parameter of the model involved in the recurrence (see Sec. 2). Most recent research on RTRL focuses on introducing *approximation methods* into the computation and storage of this matrix. For example, Menick et al. [11] introduce sparsity in both the weights of the RNN and updates of the temporal Jacobian (which is an intermediate matrix needed to compute the sensitivity matrix). Another line of work [8, 9, 10] proposes estimators based on low-rank decompositions of the sensitivity matrix that are less expensive to compute and store than the original one. Silver et al. [12] explore random projections of the sensitivity. The main research question in these lines of work is naturally focused around the quality of the proposed approximation method. Consequently, the central goal of their experiments is typically to test hyper-parameters and configurational choices that control the approximation quality in diagnostic settings, rather than evaluating the full potential of RTRL in realistic tasks. In the end, we still know very little about the true empirical promise of RTRL. Also, assuming that a solution is found to the complexity bottleneck, what actual applications or algorithms would RTRL unlock? In what scenarios would RTRL be able to replace BPTT in today's deep learning?

Here we propose to study RTRL by looking ahead beyond research on approximations. We explore the full potential of RTRL in the settings where no approximation is needed, while at the same time, not restricting ourselves to toy tasks. For that, we focus on special RNN architectures with element-wise recurrence, that allow for tractable RTRL without any approximation. In fact, the quadratic/cubic complexities of the fully recurrent NNs can be simplified for certain neural architectures. Many well-known RNN architectures, such as Quasi-RNNs [13] and Simple Recurrent Units [14], and even certain Linear Transformers [15, 16, 17], belong to this class of models (see Sec. 3.1). Note that the core idea underlying this observation is technically not new: Mozer [18, 19] already explore an RNN architecture with this property in the late 1980s to derive his *focused backpropagation*, and Javed et al. [20, 21] also exploit this in the architectural design of their RNNs (even though the problematic multi-layer case is ignored; we discuss it in Sec. 5). While such special RNNs may suffer from limited computational capabilities on certain tasks (i.e., one can come up with a synthetic/algorithmic task where such models fail; see Appendix B.1), they also often perform on par with fully recurrent NNs on many tasks (at least, this is the case for the tasks we explore in our experiments). For the purpose of this work, the RTRL-tractability property outweighs the potentially limited computational capabilities: these architectures allow us to focus on evaluating RTRL on challenging tasks with a scale that goes beyond the one typically used in prior RTRL work, and to draw conclusions without worrying about the quality of approximation. We study an actor-critic algorithm [22, 23, 24] that combines RTRL and recurrent policy gradients [25], allowing credit assignments throughout an entire episode in reinforcement learning (RL) with partially observable Markov decision processes (POMDPs; [26, 27]). We test the resulting algorithm, Real-Time Recurrent Actor-Critic method (R2AC), in several subsets of DMLab-30 [28], ProcGen [29], and Atari 2600 [30] environments, with a focus on memory tasks but also including reactive ones. In particular, on two memory environments of DMLab-30, our system is competitive with or outperforms the well-known IMPALA [31] and R2D2 [32] baselines, demonstrating certain practical benefits of RTRL at scale. Finally, working with concrete real-world tasks also sheds lights on further limitations of RTRL that are rarely (if not never) discussed in prior work. These observations are important for future research on practical RTRL. We highlight and discuss these general challenges of RTRL (Sec. 5).

## 2   Background

Here we first review real-time recurrent learning (RTRL; [1, 2, 3, 4]), which is a gradient-based learning algorithm for sequence-processing RNNs—an alternative to the now standard BPTT.

**Preliminaries.**   Let $t$, $T$, $N$, and $D$ be positive integers. We describe the corresponding learning algorithm for the following standard RNN architecture [33] that transforms an input $\boldsymbol{x}(t) \in \mathbb{R}^D$ to an output $\boldsymbol{h}(t) \in \mathbb{R}^N$ at every time step $t$ as

$$\boldsymbol{s}(t) = \boldsymbol{W}\boldsymbol{x}(t) + \boldsymbol{R}\boldsymbol{h}(t-1) \quad ; \quad \boldsymbol{h}(t) = \sigma(\boldsymbol{s}(t)) \tag{1}$$

where $\boldsymbol{W} \in \mathbb{R}^{N \times D}$ and $\boldsymbol{R} \in \mathbb{R}^{N \times N}$ are trainable parameters, $\boldsymbol{s}(t) \in \mathbb{R}^N$, and $\sigma$ denotes the element-wise sigmoid function (we omit biases). For the derivation, it is convenient to describe each

component $\boldsymbol{s}_k(t) \in \mathbb{R}$ of vector $\boldsymbol{s}(t)$ for $k \in \{1, ..., N\}$,

$$\boldsymbol{s}_k(t) = \sum_{n=1}^{D} \boldsymbol{W}_{k,n} \boldsymbol{x}_n(t) + \sum_{n=1}^{N} \boldsymbol{R}_{k,n} \sigma(\boldsymbol{s}_n(t-1)) \tag{2}$$

In addition, we consider some loss function $\mathcal{L}^{\text{total}}(1, T) = \sum_{t=1}^{T} \mathcal{L}(t) \in \mathbb{R}$ computed on an arbitrary sequence of length $T$ where $\mathcal{L}(t) \in \mathbb{R}$ is the loss at each time step $t$, which is a function of $\boldsymbol{h}(t)$ (we omit writing down explicit dependencies over the model parameters). Importantly, we assume that $\mathcal{L}(t)$ can be computed *solely* from $\boldsymbol{h}(t)$ at step $t$ (i.e., $\mathcal{L}(t)$ has no dependency on any other past activations apart from $\boldsymbol{h}(t-1)$ which is needed to compute $\boldsymbol{h}(t)$).

The role of a gradient-based learning algorithm is to efficiently compute the gradients of the loss w.r.t. the trainable parameters of the model, i.e., $\dfrac{\partial \mathcal{L}^{\text{total}}(1, T)}{\partial \boldsymbol{W}_{i,j}} \in \mathbb{R}$ for all $i \in \{1, ..., N\}$ and $j \in \{1, ..., D\}$, and $\dfrac{\partial \mathcal{L}^{\text{total}}(1, T)}{\partial \boldsymbol{R}_{i,j}} \in \mathbb{R}$ for all $i, j \in \{1, ..., N\}$. RTRL and BPTT differ in the way to compute these quantities. While we focus on RTRL here, for the sake of completeness, we also provide an analogous derivation for BPTT in Appendix A.3.

**Real-Time Recurrent Learning (RTRL).** RTRL can be derived by first decomposing the total loss $\mathcal{L}^{\text{total}}(1, T)$ over time, and then summing all derivatives of each loss component $\mathcal{L}(t)$ w.r.t. intermediate variables $\boldsymbol{s}_k(t)$ for all $k \in \{1, ..., N\}$:

$$\frac{\partial \mathcal{L}^{\text{total}}(1, T)}{\partial \boldsymbol{W}_{i,j}} = \sum_{t=1}^{T} \frac{\partial \mathcal{L}(t)}{\partial \boldsymbol{W}_{i,j}} = \sum_{t=1}^{T} \left( \sum_{k=1}^{N} \frac{\partial \mathcal{L}(t)}{\partial \boldsymbol{s}_k(t)} \times \frac{\partial \boldsymbol{s}_k(t)}{\partial \boldsymbol{W}_{i,j}} \right) \tag{3}$$

In fact, unlike BPTT that can only compute the derivative of the total loss $\mathcal{L}^{\text{total}}(1, T)$ efficiently, RTRL is an online algorithm that computes each term $\dfrac{\partial \mathcal{L}(t)}{\partial \boldsymbol{W}_{i,j}}$ through the decomposition above. The first factor $\dfrac{\partial \mathcal{L}(t)}{\partial \boldsymbol{s}_k(t)}$ can be straightforwardly computed through standard backpropagation (as stated above, we assume there is no recurrent computation between $\boldsymbol{s}(t)$ and $\mathcal{L}(t)$). For the second factor $\dfrac{\partial \boldsymbol{s}_k(t)}{\partial \boldsymbol{W}_{i,j}}$, which is an element of the so-called *sensitivity matrix/tensor*, we can derive a *forward recursion formula*, which can be obtained by directly differentiating Eq. 2:

$$\frac{\partial \boldsymbol{s}_k(t)}{\partial \boldsymbol{W}_{i,j}} = \boldsymbol{x}_j(t) \mathbb{1}_{k=i} + \sum_{n=1}^{N} \boldsymbol{R}_{k,n} \sigma'(\boldsymbol{s}_n(t-1)) \frac{\partial \boldsymbol{s}_n(t-1)}{\partial \boldsymbol{W}_{i,j}} \tag{4}$$

where $\mathbb{1}_{k=i}$ denotes the indicator function: $\mathbb{1}_{k=i} = 1$ if $k = i$, and 0 otherwise, and $\sigma'$ denotes the derivative of the sigmoid, i.e, $\sigma'(\boldsymbol{s}_n(t-1)) = \sigma(\boldsymbol{s}_n(t-1))(1 - \sigma(\boldsymbol{s}_n(t-1)))$. The derivation is similar for $\dfrac{\partial \mathcal{L}(t)}{\partial \boldsymbol{R}_{i,j}}$ where we obtain a recurrent formula to compute $\dfrac{\partial \boldsymbol{s}_k(t)}{\partial \boldsymbol{R}_{i,j}}$. As this algorithm requires to store $\dfrac{\partial \boldsymbol{s}_k(t)}{\partial \boldsymbol{W}_{i,j}}$ and $\dfrac{\partial \boldsymbol{s}_k(t)}{\partial \boldsymbol{R}_{i,j}}$, its space complexity is $O((D+N)N^2) \sim O(N^3)$. The time complexity to update the sensitivity matrix/tensor via Eq. 4 is $O(N^4)$. To be fair with BPTT, it should be noted that $O(N^4)$ is the complexity for one update; this means that the time complexity to process a sequence of length $T$ is $O(TN^4)$.

Thanks to the forward recursion, the update frequency of RTRL is flexible: one can opt for the *fully online learning*, where we update the weights using $\dfrac{\partial \mathcal{L}(t)}{\partial \boldsymbol{W}}$ at every time step, or accumulate gradients for several time steps. It should be noted that frequent updates may result in *staleness* of the sensitivity matrix, as it accumulates updates computed using old weights (Eq. 4).

Note that algorithms similar to RTRL have been derived from several independent authors (see, e.g., [3, 18], or [34, 35] for the continuous-time version).

## 3 Method

Our main algorithm is an actor-critic method that combines RTRL with recurrent policy gradients, using a special RNN architecture that allows for tractable RTRL. Here we describe its main components: an element-wise LSTM with tractable RTRL (Sec. 3.1), and the actor-critic algorithm that builds upon IMPALA [31] (Sec. 3.2).

### 3.1 RTRL for LSTM with Element-wise Recurrence (eLSTM)

The core RNN architecture we use in this work is a variant of long short-term memory (LSTM; [36]) RNN with *element-wise recurrence*. Let $\odot$ denote element-wise multiplication. At each time step $t$, it first transforms an input vector $\boldsymbol{x}(t) \in \mathbb{R}^D$ to a recurrent hidden state $\boldsymbol{c}(t) \in \mathbb{R}^N$ as follows:

$$\boldsymbol{f}(t) = \sigma(\boldsymbol{F}\boldsymbol{x}(t) + \boldsymbol{w}^f \odot \boldsymbol{c}(t-1)) \quad ; \quad \boldsymbol{z}(t) = \tanh(\boldsymbol{Z}\boldsymbol{x}(t) + \boldsymbol{w}^z \odot \boldsymbol{c}(t-1)) \tag{5}$$

$$\boldsymbol{c}(t) = \boldsymbol{f}(t) \odot \boldsymbol{c}(t-1) + (1 - \boldsymbol{f}(t)) \odot \boldsymbol{z}(t) \tag{6}$$

where $\boldsymbol{f}(t) \in \mathbb{R}^N$, $\boldsymbol{z}(t) \in \mathbb{R}^N$ are activations, $\boldsymbol{F} \in \mathbb{R}^{N \times D}$ and $\boldsymbol{Z} \in \mathbb{R}^{N \times D}$ are trainable weight matrices, and $\boldsymbol{w}^f \in \mathbb{R}^N$ and $\boldsymbol{w}^z \in \mathbb{R}^N$ are trainable weight vectors. These operations are followed by a gated feedforward NN to obtain an output $\boldsymbol{h}(t) \in \mathbb{R}^N$ as follows:

$$\boldsymbol{o}(t) = \sigma(\boldsymbol{O}\boldsymbol{x}(t) + \boldsymbol{W}^o \boldsymbol{c}(t)); \quad \boldsymbol{h}(t) = \boldsymbol{o}(t) \odot \boldsymbol{c}(t) \tag{7}$$

where $\boldsymbol{O} \in \mathbb{R}^{N \times D}$ and $\boldsymbol{W}^o \in \mathbb{R}^{N \times N}$ are trainable weight matrices. This architecture can be seen as an extension of Quasi-RNN [13] with element-wise recurrence in the gates, or Simple Recurrent Units [14] without depth gating, and also relates to IndRNN [37]. While one could further discuss myriads of architectural details [38], most of them are irrelevant to our discussion on the complexity reduction in RTRL; the only essential property here is that "recurrence" is element-wise. We use this simple architecture above, an LSTM with element-wise recurrence (or eLSTM), for all our experiments.

Furthermore, we restrict ourselves to the one-layer case (we discuss the multi-layer case later in Sec. 5), where we assume that there is no recurrence after this layer. Based on this assumption, gradients for the parameters $\boldsymbol{O}$ and $\boldsymbol{W}^o$ in Eq. 7 can be computed by the standard backpropagation, as they are not involved in recurrence. Hence, the sensitivity matrices we need for RTRL (Sec. 2) are: $\dfrac{\partial \boldsymbol{c}(t)}{\partial \boldsymbol{F}}$, $\dfrac{\partial \boldsymbol{c}(t)}{\partial \boldsymbol{Z}} \in \mathbb{R}^{N \times N \times N}$, and $\dfrac{\partial \boldsymbol{c}(t)}{\partial \boldsymbol{w}^f}, \dfrac{\partial \boldsymbol{c}(t)}{\partial \boldsymbol{w}^z} \in \mathbb{R}^{N \times N}$. Through trivial derivations, we can show that each of these sensitivity matrices can be computed using a tractable forward recursion formula (we provide the full derivation in Appendix A.1). For example for $\dfrac{\partial \boldsymbol{c}(t)}{\partial \boldsymbol{F}}$, we have, for $i, j, k \in \{1, ..., N\}$,

$$\hat{\boldsymbol{f}}_i(t) = (\boldsymbol{c}_i(t-1) - \boldsymbol{z}_i(t))\boldsymbol{f}_i(t)(1 - \boldsymbol{f}_i(t)) \tag{8}$$

$$\frac{\partial \boldsymbol{c}_i(t)}{\partial \boldsymbol{F}_{i,j}} = (\boldsymbol{f}_i(t) + \boldsymbol{w}_i^f \hat{\boldsymbol{f}}_i(t))\frac{\partial \boldsymbol{c}_i(t-1)}{\partial \boldsymbol{F}_{i,j}} + \hat{\boldsymbol{f}}_i(t)\boldsymbol{x}_j(t) \, ; \text{ and } \quad \frac{\partial \boldsymbol{c}_k(t)}{\partial \boldsymbol{F}_{i,j}} = 0 \quad \text{for all } k \neq i. \tag{9}$$

where we introduce an intermediate vector $\hat{\boldsymbol{f}}(t) \in \mathbb{R}^N$ with components $\hat{\boldsymbol{f}}_i(t) \in \mathbb{R}$ for convenience. Consequently, the gradients for the weights can be computed as:

$$\frac{\partial \mathcal{L}(t)}{\partial \boldsymbol{F}_{i,j}} = \sum_{k=1}^{N} \frac{\partial \mathcal{L}(t)}{\partial \boldsymbol{c}_k(t)} \times \frac{\partial \boldsymbol{c}_k(t)}{\partial \boldsymbol{F}_{i,j}} = \frac{\partial \mathcal{L}(t)}{\partial \boldsymbol{c}_i(t)} \times \frac{\partial \boldsymbol{c}_i(t)}{\partial \boldsymbol{F}_{i,j}} \tag{10}$$

**Finally**, we can compactly summarise these equations using the standard matrix operations. By introducing notations $\hat{\boldsymbol{F}}(t) \in \mathbb{R}^{N \times N}$ with $\hat{\boldsymbol{F}}_{i,j}(t) = \dfrac{\partial \boldsymbol{c}_i(t)}{\partial \boldsymbol{F}_{i,j}} \in \mathbb{R}$, and $\boldsymbol{e}(t) \in \mathbb{R}^N$ with $\boldsymbol{e}_i(t) = \dfrac{\partial \mathcal{L}(t)}{\partial \boldsymbol{c}_i(t)} \in \mathbb{R}$ for $i \in \{1, ..., N\}$ and $j \in \{1, ..., D\}$, Eqs. 8-10 above can be written as:

$$\hat{\boldsymbol{f}}(t) = (\boldsymbol{c}(t-1) - \boldsymbol{z}(t)) \odot \boldsymbol{f}(t) \odot (1 - \boldsymbol{f}(t)) \tag{11}$$

$$\hat{\boldsymbol{F}}(t) = \text{diag}\left(\boldsymbol{f}(t) + \boldsymbol{w}^f \odot \hat{\boldsymbol{f}}(t)\right)\hat{\boldsymbol{F}}(t-1) + \hat{\boldsymbol{f}}(t) \otimes \boldsymbol{x}(t) \quad ; \quad \frac{\partial \mathcal{L}(t)}{\partial \boldsymbol{F}} = \text{diag}(\boldsymbol{e}(t))\hat{\boldsymbol{F}}(t) \tag{12}$$

where, for notational convenience, we introduce a function $\text{diag} : \mathbb{R}^N \to \mathbb{R}^{N \times N}$ that constructs a diagonal matrix whose diagonal elements are those of the input vector; however, in practical implementations (e.g., in PyTorch), this can be directly handled as vector-matrix multiplications with broadcasting (this is an important note for complexity analysis). $\otimes$ denotes outer-product.

Analogously, we can derive compact update equations of sensitivity matrices and gradient computations for other parameters $\boldsymbol{Z}$, $\boldsymbol{w}^f$ and $\boldsymbol{w}^z$ (as well as biases which are omitted here). The complete list of these equations is provided in Appendix A.1.

The RTRL algorithm above requires maintaining sensitivity matrices $\hat{\boldsymbol{F}}(t) \in \mathbb{R}^{N \times N}$, and analogously defined $\hat{\boldsymbol{Z}}(t) \in \mathbb{R}^{N \times N}$, $\hat{\boldsymbol{w}}^f(t) \in \mathbb{R}^N$, and $\hat{\boldsymbol{w}}^z(t) \in \mathbb{R}^N$ (see Appendix A.1); thus, the space complexity is $O(N^2)$. The per-step time complexity is $O(N^2)$ (see Eqs. 8-10). This is all tractable. Importantly, these equations 11-12 can be implemented as simple PyTorch code (just like the forward pass of the same model; Eqs. 5-7) without any non-standard logics. Note that many approximations of RTRL often involve computations that are not well supported yet in the standard deep learning library (e.g., efficiently handling custom sparsity), which is an extra barrier for scaling RTRL in practice.

Note that the derivation of RTRL for element-wise recurrent nets is not novel: similar methods can be found in Mozer [18, 19] from the late 1980s. This result itself is also not very surprising, since element-wise recurrence introduces obvious sparsity in the temporal Jacobian (which is part of the second term in Eq. 4). Nethertheless, we are not aware of any prior work pointing out that several modern RNN architectures such Quasi-RNN [13] or Simple Recurrent Units [14] yield tractable RTRL (in the one-layer case). Also, while this is not the focus of our experiments, we show an example of Linear Transformers/Fast Weight Programmers [15, 16, 17] that have tractable RTRL (details can be found in Appendix A.2), which is another conceptually interesting result. We also note that the famous LSTM-algorithm [36] (companion learning algorithm for the LSTM architecture) is a diagonal approximation of RTRL, so is the more recent SnAp-1 of Menick et al. [11]. Unlike in these works, the gradients computed by our RTRL algorithm above are *exact* for our eLSTM architecture. This allows us to draw conclusions from experimental results without worrying about the potential influence of approximation quality. We can evaluate the full potential of RTRL for this specific architecture.

Finally, this is also an interesting system from the biological standpoint. Each weight in the weight matrix/synaptic connections (e.g., $\boldsymbol{F} \in \mathbb{R}^{N \times N}$) is augmented with the corresponding "memory" ($\hat{\boldsymbol{F}}(t) \in \mathbb{R}^{N \times N}$) tied to its own learning process, which is updated in an online fashion, as the model observes more and more examples, through an Hebbian/outer product-based update rule (Eq. 12/Left).

## 3.2 Real-Time Recurrent Actor-Critic Policy Gradient Algorithm (R2AC)

The main algorithm we study in this work, Real-Time Recurrent Actor-Critic method (R2AC), combines RTRL with recurrent policy gradients. Our algorithm builds upon IMPALA [31]. Essentially, we replace the RNN archicture and its learning algorithm, LSTM/TBPTT in the standard recurrent IMPALA algorithm, by our eLSTM/RTRL (Sec. 3.1). While we refer to the original paper [31] for basic details of IMPALA, here we recapitulate some crucial aspects. Let $M$ denote a positive integer. IMPALA is a distributed actor-critic algorithm where each *actor* interacts with the environment for a fixed number of steps $M$ to obtain a state-action-reward trajectory segment of length $M$ to be used by the *learner* to update the model parameters. $M$ is an important hyper-parameter that is used to specify the number of steps $M$ for $M$-step TD learning [39] of the critic, and the frequency of weight updates. Given the same number of environmental steps used for training, systems trained with a smaller $M$ apply more weight updates than those trained with a higher $M$. For recurrent policies trained with TBPTT, $M$ also represents the BPTT span (i.e., BPTT is carried out on the $M$-length trajectory segment; no gradient is propagated farther than $M$ steps back in time; while the last state of the previous segment is used as the initial state of the new segment in the forward pass). In the case of RTRL, there is no gradient truncation, but since $M$ controls the update frequency, the greater the $M$, the less frequently we update the parameters, and it potentially suffers less from sensitivity matrix staling. This setting allows for comparing TBPTT and RTRL in the setting where everything is equal (including the number of updates) except the actual gradients applied to the weights: truncated vs. untruncated ones.

Note that for R2AC with $M = 1$, one could obtain a *fully online* recurrent actor-critic method. However, in practice, it is known that $M > 1$ is crucial (for TD learning of the critic) for optimal performance. In all our experiments, we have $M > 1$. The main focus of this work is to evaluate learning with untruncated gradients, rather than the potential for online learning.

## 4 Experiments

### 4.1 Diagnostic Task

Since the main focus of this work is to evaluate RTRL-based algorithms beyond diagnostic tasks, we only conduct brief experiments on a classic diagnostic task used in recent RTRL research work focused on approximation methods [8, 9, 10, 11, 12]: the copy task. Since our RTRL algorithm (Sec. 3.1) requires no approximation, and the task is trivial, we achieve 100% accuracy provided that the RNN size is large enough and that training hyper-parameters are properly chosen. We confirm this for sequences with lengths of up to 1000. Additional experimental details can be found in Appendix B.1.

### 4.2 Memory Tasks

Here we present the main experiments of this work: RL in POMDPs using realistic game environments requiring memory.

**DMLab Memory Tasks.** DMLab-30 [28] is a collection of 30 first-person 3D game environments, with a mix of both memory and reactive tasks. Here we focus on two well-known environments, `rooms_select_nonmatching_object` and `rooms_watermaze`, which are both categorised as "memory" tasks according to Parisotto et al. [40]. The mean episode lengths of these tasks are about 100 and 1000 steps, respectively. As we apply an action repetition of 4, each "step" corresponds to 4 environmental frames here. We refer to Appendix B.2 for further descriptions of these tasks, and experimental details. Our model architecture is based on that of IMPALA [31]. Both RTRL and TBPTT systems use our eLSTM (Sec. 3.1) as the recurrent layer with a hidden state size of 512. Everything is equal between these two systems except that the gradients are truncated in TBPTT but not in RTRL. To reduce the overall compute needed for the experiments, we first pre-train one TBPTT model for 50 M steps for `rooms_select_nonmatching_object`, and for 200 M steps for `rooms_watermaze`. Then, for all main training runs in this experiment, we initialise the parameters of the convolutional vision module from the same pre-trained model, and keep these parameters frozen (and thus, only train the recurrent layer and everything above it). For these main training runs, we train for 30 M and 100 M steps for `rooms_select_nonmatching_object` and `rooms_watermaze`, respectively; resulting in the total of 320 M and 1.2 B environmental frames. We compare RTRL and TBPTT for different values of $M \in \{10, 50, 100\}$ (Sec. 3.2). We recall that $M$ influences: the frequency of weight updates, $M$-step TD learning, as well as the backpropagation span for TBPTT.

Table 1 shows the corresponding scores, and the left part of Figure 1 shows the training curves. We observe that for `select_nonmatching_object` which has a short mean episode length of 100 steps, the performance of TBPTT and RTRL is similar even with $M = 50$. The benefit of RTRL is only visible in the case with $M = 10$. In contrast, for the more challenging `rooms_watermaze` task with a mean episode length of 1000 steps, RTRL outperforms TBPTT for all values of $M \in \{10, 50, 100\}$. Furthermore, with $M = 50$ or 100, our RTRL system outperforms the IMPALA and R2D2 systems from prior work [32], while trained on fewer than 1.2 B frames. Note that R2D2 systems [32] are trained without action repetitions, and with a BPTT span of 80. This effectively demonstrates the practical benefit of RTRL in a realistic task requiring long-span credit assignments.

**ProcGen.** We test R2AC in another domain: ProcGen [29]. Most ProcGen environments are solvable using a feedforward policy even without frame-stacking [29]. There is a so-called *memory-mode* for certain games, making the task partially observable by making the world bigger, and restricting agents' observations to a limited area around them. However, in our preliminary experiments, we observe that even in these POMDP settings, both the feedforward and LSTM baselines perform similarly (see Appendix B.3). Nevertheless, we find one environment in the standard *hard-mode*, *Chaser*, which shows clear benefits of recurrent policies over those without memory. *Chaser* is similar to the classic game "Pacman," effectively requiring some counting capabilities to fully exploit *power pellets* valid for a limited time span. The mean episode length for this task is about 200 steps, where each step is an environmental frame as we apply no action repeat for ProcGen. Unlike in the DMLab experiments above, here we train all models from scratch for 200 M steps without pre-training the vision module (since training the vision parameters using RTRL is intractable, they are trained with truncated gradients, i.e., only the recurrent layer is trained using RTRL; we discuss this further in Sec. 5). We compare RTRL and TBPTT with $M = 5$ or 50. The training curves are shown in the right part of Figure 1.

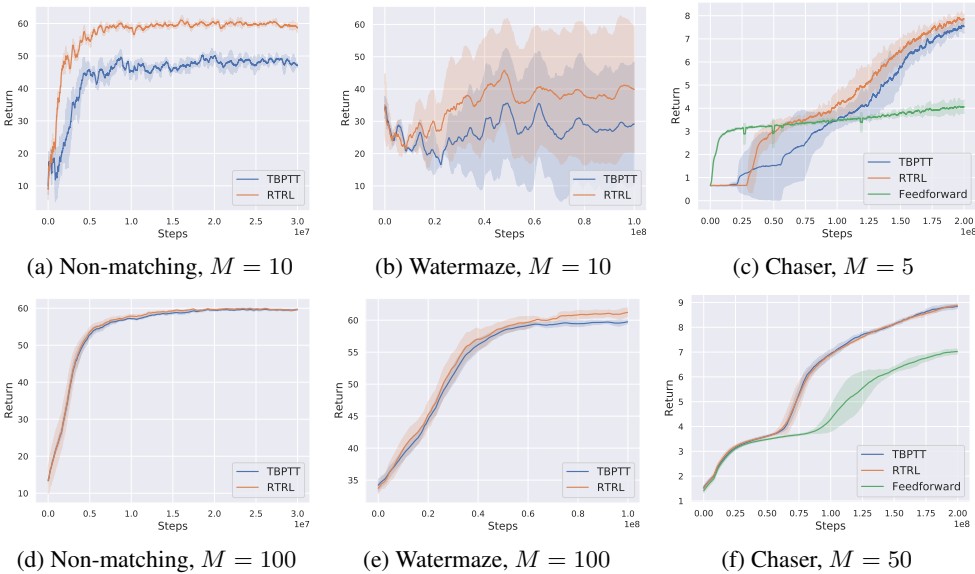

|  |  |  |  |  |  |
| --- | --- | --- | --- | --- | --- |
| (a) Non-matching, $M = 10$ | | (b) Watermaze, $M = 10$ | | (c) Chaser, $M = 5$ | |
| (d) Non-matching, $M = 100$ | | (e) Watermaze, $M = 100$ | | (f) Chaser, $M = 50$ | |

Figure 1: Training curves on **DMLab-30** `rooms_select_nonmatching_object` (Non-matching) and `rooms_watermaze` (Watermaze), and **Procgen** *Chaser* environments.

Table 1: Final game scores on two memory environments of **DMLab-30**: `rooms_select_nonmatching_object` and `rooms_watermaze`. Numbers on the top part are copied from the respective papers for reference. We report mean and standard deviation computed over 3 training seeds (each using 3 sets of 100 test episodes; see Appendix B.2). "frames" indicates the number of environmental frames used for training. $M$ is the hyper-parameter that controls weight update frequency, $M$-step TD learning, and backpropagation span for TBPTT in IMPALA (see Sec. 3.2).

|  | frames | $M$ | select_nonmatching_object | watermaze |
| --- | --- | --- | --- | --- |
| IMPALA ([31]) | 1 B | 100 | 7.3 | 26.9 |
| IMPALA ([32]) | 10 B | 100 | 39.0 | 47.0 |
| R2D2 ([32]) | 10 B | - | 2.3 | 45.9 |
| R2D2+ ([32]) | 10 B | - | 63.6 | 49.0 |
| TBPTT | < 1.2B | 10 | $54.5 \pm 1.1$ | $15.8 \pm 0.9$ |
| RTRL | | | $\mathbf{61.8 \pm 0.5}$ | $\mathbf{40.2 \pm 5.6}$ |
| TBPTT | < 1.2B | 50 | $61.4 \pm 0.5$ | $44.5 \pm 1.5$ |
| RTRL | | | $\mathbf{62.0 \pm 0.4}$ | $\mathbf{52.3 \pm 1.9}$ |
| TBPTT | < 1.2B | 100 | $61.7 \pm 0.1$ | $45.6 \pm 4.7$ |
| RTRL | | | $\mathbf{62.2 \pm 0.3}$ | $\mathbf{54.8 \pm 4.3}$ |

Similar to the `rooms_select_nonmatching_object` case above, with a sufficiently large $M = 50$, there is no difference between RTRL and TBPTT, while we observe benefits of RTRL when $M = 5$.

### 4.3 General Evaluation

Here we evaluate R2AC more broadly, including environments which are mostly reactive.

**Atari.** Apart from some exceptions (such as *Solaris* [32]), many of the Atari game environments are considered to be fully observable when observations consist of a stack of 4 frames [41, 42]. However, it is also empirically known that, for certain games, recurrent policies yield higher performance than the feedforward ones having only access to 4 past frames (see, e.g., [43, 32, 44]). Here our general goal is to compare RTRL to TBPTT more broadly. We use five Atari environments: *Breakout*,

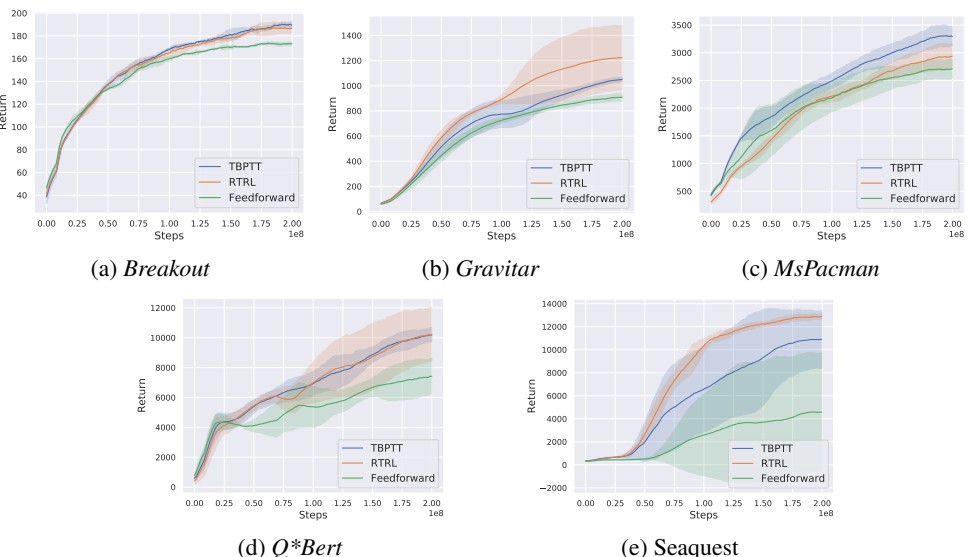

Figure 2: Learning curves on five **Atari** environments

Table 2: Scores on Atari and DMLab-reactive (`rooms_keys_doors_puzzle`) environments.

| | Breakout | Gravitar | MsPacman | Q*bert | Seaquest | `keys_doors` |
|---|---|---|---|---|---|---|
| Feedforward | $234 \pm 12$ | $1084 \pm 54$ | $3020 \pm 305$ | $7746 \pm 1356$ | $4640 \pm 3998$ | $\mathbf{26.6} \pm 1.1$ |
| TBPTT | $\mathbf{305} \pm 29$ | $1269 \pm 11$ | $\mathbf{3953} \pm 497$ | $11298 \pm 615$ | $12401 \pm 1694$ | $26.1 \pm 0.4$ |
| RTRL | $275 \pm 53$ | $\mathbf{1670} \pm 358$ | $3346 \pm 442$ | $\mathbf{12484} \pm 1524$ | $\mathbf{12862} \pm 961$ | $26.1 \pm 0.9$ |

*Gravitar*, *MsPacman*, *Q\*bert*, and *Seaquest*, following Kapturowski et al. [32]'s selection for ablations of their R2D2. Here we use $M = 50$, and train for 200 M steps (with the action repeat of 4) from scratch. The learning curves are shown in Figure 2. With the exception of *MsPacman* (note that, unlike ProcGen/*Chaser* above, 4-frame stacking is used) where we observe a slight performance degradation, RTRL performs equally well or better than TBPTT in all other environments.

**DMLab Reactive Task.** Finally, we also test our system on one environment of DMLab-30, `room_keys_doors_puzzle`, which is categorised as a reactive task according to Parisotto et al. [40]. We train with $M = 100$ for 100 M steps (with the action repeat of 4). The mean episode length is about 450 steps. Table 2/right shows the scores. Effectively, all feedforward, TBPTT, and RTRL systems perform nearly the same (at least within 100 M steps/400 M frames). We note that these scores are comparable to the one reported by the original IMPALA [31] which is 28.0 after training on 1 B frames, which is much worse than the score reported by Kapturowski et al. [32] for IMPALA trained using 10 B frames (54.6). We show this example to confirm that RTRL is effectively not helpful on a reactive task, unlike in the memory tasks above.

## 5 Limitations and Discussion

Here we discuss limitations of this work, which also sheds light on more general challenges of RTRL.

**Multi-layer case of our RTRL.** The most crucial limitation of our tractable-RTRL algorithm for element-wise recurrent nets (Sec. 3.1) is its restriction to the one-layer case. By stacking two such layers, the corresponding RTRL algorithm becomes intractable as we end up with the same complexity bottleneck as in fully recurrent networks. This is simply because by composing two such element-wise recurrent layers, we obtain a fully recurrent NN as a whole. This can be easily seen by writing down the following equations. By introducing extra superscripts to denote the layer number, in a stack of two element-wise LSTM layers of Eqs. 5-6 (we remove the output gate), we can express

the recurrent state $\boldsymbol{c}^{(2)}(t)$ of the second layer at step $t$ as a function of the recurrent state $\boldsymbol{c}^{(1)}(t-1)$ of the first layer from the previous step as follows:

$$\boldsymbol{c}^{(2)}(t) = \boldsymbol{f}^{(2)}(t) \odot \boldsymbol{c}^{(2)}(t-1) + (1 - \boldsymbol{f}^{(2)}(t)) \odot \boldsymbol{z}^{(2)}(t) \tag{13}$$

$$\boldsymbol{f}^{(2)}(t) = \sigma(\boldsymbol{F}^{(2)}\boldsymbol{c}^{(1)}(t) + \boldsymbol{w}^{f(2)} \odot \boldsymbol{c}^{(2)}(t-1)) \tag{14}$$

$$= \sigma(\boldsymbol{F}^{(2)}\left(\boldsymbol{f}^{(1)}(t) \odot \boldsymbol{c}^{(1)}(t-1) + (1 - \boldsymbol{f}^{(1)}(t)) \odot \boldsymbol{z}^{(1)}(t)\right) + ...) \tag{15}$$

$$= \sigma(\boldsymbol{F}^{(2)}\boldsymbol{f}^{(1)}(t) \odot \boldsymbol{c}^{(1)}(t-1) + \boldsymbol{F}^{(2)}(1 - \boldsymbol{f}^{(1)}(t)) \odot \boldsymbol{z}^{(1)}(t) + ...) \tag{16}$$

By looking at the first term of Eq. 13 and that of Eq. 16, one can see that there is full recurrence between $\boldsymbol{c}^{(2)}(t)$ and $\boldsymbol{c}^{(1)}(t-1)$ via $\boldsymbol{F}^{(2)}$, which brings back the quadratic/cubic time and space complexity for the sensitivity of the recurrent state in the second layer w.r.t. parameters of the first layers. This limitation is not discussed in prior work [18, 19].

**Complexity of multi-layer RTRL in general.** Generally speaking, RTRL for the multi-layer case is rarely discussed (except Meert and Ludik [45]; 1997). This case is important in modern deep learning where stacking multiple layers is a standard. There are two important remarks to be made here.

First of all, even in an NN with a single RNN layer, if there is a layer with trainable parameters whose output is connected to the input of the RNN layer, a sensitivity matrix needs to be computed and stored for each of these parameters. A good illustration is the policy net used in all our RL experiments where our eLSTM layer takes the output of a deep (feedforward) convolutional net (the vision stem) as input. As training this vision stem using RTRL requires dealing with the corresponding sensitivity matrix, which is intractable, we train/pretrain the vision stem using TBPTT (Sec. 4.2;4.3). This is an important remark for RTRL research in general. For example, approximation methods proposed for the single-layer case may not scale to the multi-layer case; e.g., to exploit sparsity in the policy net above, it is not enough to assume weight sparsity in the RNN layer, but also in the vision stem.

Second, the multi-layer case [45] introduces more complexity growth to RTRL than to BPTT. Let $L$ denote the number of layers. We seemlessly use BPTT with deep NNs, as its time and space complexity is linear in $L$. This is not the case for RTRL. With RTRL, for each recurrent layer, we need to store sensitivities of all parameters of all preceding layers. This implies that, for an $L$-layer RNN, parameters in the first layer require $L$ sensitivity matrices, $L-1$ for the second layer, ..., etc., resulting in $L + (L-1) + (L-2) + ... + 2 + 1 = L(L+1)/2$ sensitivity matrices to be computed and stored. Given that multi-layer NNs are crucial today, this remains a big challenge for practical RTRL research.

**Principled vs. practical solution.** Another important aspect of RTRL research is that many realistic memory tasks have actual dependencies/credit assignment paths that are shorter than the maximum BPTT span we can afford in practice. In our experiments, with the exception of DMLab `rooms_watermaze` (Sec. 4.2), no task actually absolutely *requires* RTRL in practice; TBPTT with a large span suffices. Future improvements of the hardware may give a further advantage to TBPTT; the *practical* (simple) solution offered by TBPTT might be prioritised over the *principled* (complex) RTRL solution for dealing with long-span credit assignments. This is also somewhat reminiscent of the Transformer vs. RNN discussion regarding sequence processing with limited vs. unlimited context.

**Sequence-level parallelism.** While our study focuses on evaluation of untruncated gradients, another potential benefit of RTRL is online learning. For most standard self/supervised sequence-processing tasks such as language modelling, however, modern implementations are optimised to exploit access to the "full" sequence, and to leverage parallel computation across the time axis (at least for training). While some hybrid RTRL-BPTT approaches [46] may still be able to exploit such a parallelism, fast online learning remains open engineering challenge even with tractable RTRL.

## 6   Conclusion

We demonstrate the empirical promise of RTRL in realistic settings. By focusing on RNNs with element-wise recurrence, we obtain tractable RTRL without approximation. We evaluate our reinforcement learning RTRL-based actor-critic in several popular game environments. In one of the challenging DMLab-30 memory environments, our system outperforms the well-known IMPALA and R2D2 baselines which use many more environmental steps. We also highlight general important limitations and further challenges of RTRL rarely discussed in prior work.

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
