# OpenReview forum: "Exploring the Promise and Limits of Real-Time Recurrent Learning"
_NeurIPS.cc/2023/Conference — Submitted to NeurIPS 2023_

### Official Review · Reviewer_Bqpj · 2023-06-30

**Soundness:** 3 good
**Presentation:** 4 excellent
**Contribution:** 3 good
**Rating:** 7
**Confidence:** 3

**Summary:**

The authors investigate Realtime Recurrent Learning (RTRL) applied to recurrent RL, in contrast with traditional Backpropagation Through Time (BPTT). The authors' key insight is that they can significantly reduce time and space complexity by using single-layer elementwise recurrence. They propose an elementwise-LSTM (eLSTM), similar to IndRNN, for this purpose.

The authors execute a toy diagnostic task, then use IMPALA and R2D2 to train two variants of their eLSTM on pixels tasks -- one variant using BPTT and the other using RTRL.

As expected, the authors find that increasing the TBPTT length to large values provides the same performance as RTRL. Their is a notable performance gap when using short TBPTT, motivating the use of their approach. Although the tasks here do not necessarily require RTRL, one can imagine POMDPs that span hundreds of thousands of timesteps where this could be useful.

**Strengths:**

- The authors present an interesting and forward-looking idea. Although we are not aware of any common RL tasks today that require such long-term backpropgatation, it's clear that human-level decision making happens over timeframes much too long for BPTT to be tractable and for TBPTT to capture such dependencies. We suspect methods like RTRL will be necessary for truly intelligent agents.
- The paper is generally well written, and it appears that the authors have dilligently reviewed prior literature.

**Weaknesses:**

- My biggest complaint is that there is no data showing empirical time or space efficiency of RTRL. My understanding is that the whole point of RTRL is decoupling training space efficiency from the sequence length. I do not expect RTRL to be faster for such short sequences, but I would like to compare practical wall-clock time or GPU memory usage.

- I think it should be made explicit that in the traditional RL scenario (rollout workers separate from trainer, sync weights from trainer to workers every update), BPTT and RTRL are equivalent. It is really only the BPTT truncation that causes issues.

- The paper could benefit from a mini-study further examining the effects of TBPTT vs BPTT. The watermaze/chaser experiments do this, but only for a very limited number of truncation lengths. It would be interesting to see results with truncation lengths at 10, 20, ... 200.

**Questions:**

- Line 24: to cache -> caching
- Diagnostic task is run for a maximum length of 1000
    - This is achievable using untruncated BPTT. Since BPTT scales with sequence length and RTRL does not, it would make more sense to do a very large sequence length.
- It is not clear to me why varying $M$ matters for RTRL if training is not "online" in the sense that the rollout workers are not updating weights

**Limitations:**

- RTRL is relatively understudied and thus has numerous limitations in its current state. As it receives further attention, I suspect such limitations will be mitigated. It is nice to see the authors be honest about the limitations of RTRL throughout the paper.

---

> ### Author Rebuttal · Authors · 2023-08-09
>
> We thank the reviewer for spending valuable time on reviewing our work. We also thank the reviewer for many positive and encouraging comments supporting the core philosophy underlying our work.
> Please find our response as follows.
>
> > *"My biggest complaint is that there is no data showing empirical time or space efficiency of RTRL. My understanding is that the whole point of RTRL is decoupling training space efficiency from the sequence length. I do not expect RTRL to be faster for such short sequences, but I would like to compare practical wall-clock time or GPU memory usage."*
>
> We first would like to clarify that our code is not optimised to fully leverage the computational advantages of RTRL.
> Nevertheless, let us try to provide some insightful numbers as follows. Training an eLSTM-RTRL with hidden size N=512, batch size 32 and M=100, using the frozen vision module on Watermaze requires about 5 GB GPU memory (corresponding to our experimental setting of Table 1). This is comparable to that of TBPTT for M=100; in fact the memory requirements of TBPTT for M=10, 50, 100, 300 are respectively: 1.4, 2.9, 4.8, 12 GB (the corresponding empirical law is 0.038 * M + 1 GB). Therefore, in order to compute the untruncated gradients as in RTRL using TBPTT for Watermaze (let’s simply assume that M=1000 is needed as 1000 is the average episode length for Watermaze; Line 217), the corresponding memory requirement is greater than 39 GB. In contrast, RTRL using only 5 GB allows to compute untruncated gradients and yields good performance (Table 1).
>
> Regarding the actual wall clock time (measured on V100), TBPTT/PyTorch-backward/automatic-differentiation is a bit faster (1900 steps per second) than our RTRL/custom-forward-differentiation (1250 steps per second) but again, our implementation is not professionally optimised, and leaves much room for improvements (formally, the time complexity of O($MN^2$) is the same for TBPTT and RTRL applied to our eLSTM, but RTRL requires some extra overheads such as copying large sensitivity matrices as extra recurrent states).
>
> > *"I think it should be made explicit that in the traditional RL scenario (rollout workers separate from trainer, sync weights from trainer to workers every update), BPTT and RTRL are equivalent. It is really only the BPTT truncation that causes issues."*
>
> Thank you for pointing this out. In fact, we do already state this in the background section (Line 92-95 *“RTRL and BPTT differ in the way to compute these quantities.“*), but given that Reviewer 5hxy is also asking for experiments to report results for full BPTT (which seems to indicate that this equivalence was not clear), we will stress more on this in the final version.
>
> > *"The paper could benefit from a mini-study further examining the effects of TBPTT vs BPTT. The watermaze/chaser experiments do this, but only for a very limited number of truncation lengths. It would be interesting to see results with truncation lengths at 10, 20, ... 200."*
>
> We’d like to draw the reviewer’s attention to the fact that such a *“mini-study further examining the effects of TBPTT vs BPTT”* is already well illustrated in the case of DMLab Select-Non-Matching-Object (Table 1). There, with M=10, RTRL clearly outperforms TBPTT, but by increasing M to up to 100, RTRL and TBPTT perform similarly.
> In fact, the average episode length for this task is about 100 (as we state in this in Line 217), and therefore, the M=100 case almost corresponds to the full-BPTT limit case for TBPTT, which is mathematically equivalent to RTRL in terms of computed gradients.
> If the reviewer still thinks that finer ticks on M may bring other important observations (if so please explain why), we’d be happy to run more experiments for the final version, but otherwise, we’d like to argue that the current experiments are already illustrating the corresponding effect sufficiently well.
>
> > *"Line 24: to cache -> caching"*
>
> Thank you for pointing this out. We will fix this in the final version.
>
> > *"Diagnostic task is run for a maximum length of 1000"*
>
> Thank you for pointing this out. In fact, the diagnostic task is there really just for the diagnostic purpose. Therefore, we did not try further experiments on this task, and instead, we focused on more realistic and interesting RL experiments.
>
> > *"It is not clear to me why varying M matters for RTRL if training is not "online" in the sense that the rollout workers are not updating weights"*
>
> This is explained in Line 194 *“In the case of RTRL, …”*.
> M determines the “frequency” of weight updates for both RTRL and TBPTT (i.e., we update the weights once every "M times batch-size" frames). Therefore, for RTRL, M influences “staleness” of the sensitivity matrices (explained in Line 115 in the background section): greater M implies less staling (which is good). This way, even if the gradients are untruncated for any M, the actual gradients computed by RTRL are different for different values of M. If the reviewer still finds this confusing, we'll be happy to try to improve the corresponding text.
>
> We hope our response brings clarifications to all of the reviewer’s remaining questions. If you find our response useful, and/or if you think our paper should be accepted, please consider increasing the scores. Thank you.

---

> > ### Comment · Reviewer_Bqpj · 2023-08-11
> >
> > Thank you for the response and clarification. That is much faster and more efficient than I anticipated. It sounds like RTRL could soon be more practical than BPTT. I'm looking forward to seeing RTRL enable online updates and what challenges that will bring.

---

> > > ### Author Response · Authors · 2023-08-11
> > > **Thank you for your response**
> > >
> > > Thank you very much for your response!
> > >
> > > > *"RTRL could soon be more practical than BPTT."*
> > >
> > > Unfortunately, we have to admit that this is likely not the case yet, due to all the open challenges and limitations of RTRL in more general cases (which, as the reviewer has already commented, we critically and honestly discussed in the paper; while never discussed in prior work).
> > >
> > > However, we hope we managed to show the promise of RTRL applied to challenging RL problems, at least in the very restricted case of 1-layer element-wise RNNs (eLSTM), where we could evaluate RTRL without worrying about the above limitations.
> > >
> > > We are glad to see that the reviewer seems to find this promising overall!
> > >
> > > If you found our response useful, and/or if you think our paper should be accepted, please do not hesitate to increase your score.
> > > Thank you very much.

---

### Official Review · Reviewer_Y8rh · 2023-07-05

**Soundness:** 4 excellent
**Presentation:** 4 excellent
**Contribution:** 3 good
**Rating:** 7
**Confidence:** 3

**Summary:**

This paper explores an alternative to back-propagation through time (BPTT) for training recurrent neural networks, called real time recurrent learning (RTRL). The alternative algorithm does not depend on past activations, and also does not need truncation (which limits past context) like in BPTT to make it tractable. Although RTRL is very computationally complex for the general case, this paper exploits the fact that several recent recurrent architectures (Quasi-RNNs) allow for exact RTRL gradients to be computed efficiently. The method is evaluated on reinforcement learning tasks on several benchmarks and is compared with truncated BPTT  with LSTM while using Actor-Critic method as RL algorithms.

**Strengths:**

1.	The paper studies a very interesting direction of using RTRL instead of truncated BPTT. The difficulty of using BPTT for training recurrent architectures is well known and it might be a potential bottleneck in allowing recurrent networks in achieving their full performance in complex tasks in the real-world.
2.	The paper is written in a very clear way, clearly outlining the core contributions and the focus of this work.
3.	The proposed method is very clean, elegant, well-explained, and simple.
4.	The empirical evaluation is done very extensively on DMLab, ProcGen and Atari, and it's interesting to see that RTRL + A2C performs equivalent, or better than TBPTT, except for in a few scenarios.
5.	The limitations section is well-written and shows a good understanding of the proposed technique.


**Weaknesses:**

Weaknesses / Questions:
1. Ablate the dependence of M (episode rollout length) on R2AC. Since larger M leads to less frequent updates and shorter M leads to stale values in the sensitivity matrix, it would be interesting to see how the performance varies with this parameter.
2. One modification to the empirical evaluation that I would suggest is to compare with LSTM (instead of eLSTM), and also to compare the best performance for both the training algorithms – TBPTT and R2AC for any value of M, tuned separately for the two algorithms.
3. How does BPTT + LSTM with multiple hidden layers perform in the considered benchmarks?


**Questions:**

(see weaknesses section for questions)



**Limitations:**

The limitations section is well-written and gives a clear explanation of the limits of the proposed line of work to train recurrent networks.

---

> ### Author Rebuttal · Authors · 2023-08-09
>
> We thank the reviewer for spending valuable time on reviewing our work and for many positive and supportive comments.
> Please find our response as follows:
>
> > *"Ablate the dependence of M (episode rollout length) on R2AC."*
>
> > *"… also to compare the best performance for both the training algorithms – TBPTT and R2AC for any value of M, tuned separately for the two algorithms."*
>
> We’d like to draw the reviewer’s attention to the fact that Table 1 already shows certain ablation studies on M. Overall, increasing M tends to help in our settings (indicating the importance of the backpropagation span for TBPTT and the sensitivity’s freshness for RTRL).
> Based on this observation, the “best” value for TBPTT might require increasing M until it covers the entire episode length, converging towards full BPTT is equivalent to RTRL (mathematically they compute exactly the same gradients).
> Instead, here our main focus is to show the benefits of RTRL/untruncated gradients within the range of M (typically used in prior work) for which TBPTT remains effectively truncated.
>
> As a side note, the case of DMLab Select-Non-Matching-Object (Table 1) is a good illustration of the limit case described above where both TBPTT and RTRL performances staturate and become very similar (see M=50 and M=100); this is the case because the mean episode length for this task is about 100 (as we note in Line 217), so TBPTT with M=100 is almost equal to full BPTT here.
>
> > *"One modification to the empirical evaluation that I would suggest is to compare with LSTM (instead of eLSTM)"*
>
> > *"How does BPTT + LSTM with multiple hidden layers perform in the considered benchmarks?"*
>
> As requested by the reviewer, we conducted extra experiments.
> Please refer to our general response for the results.
>
> We hope our response and extra experimental results bring clarifications to all of the reviewer’s questions. If you find our response useful, and/or if you think our paper should be accepted, please consider increasing the scores. Thank you.

---

### Official Review · Reviewer_5hxy · 2023-07-05

**Soundness:** 3 good
**Presentation:** 3 good
**Contribution:** 3 good
**Rating:** 4
**Confidence:** 5

**Summary:**

- The paper explores the use of RTRL in the case of scenario, where no approximation is needed.
- The proposed "method" is not novel per se i.e., Mozer et. al put forward the derivation of RTRL for element-wise recurrence, but to the best of reviewer's ability no one has formally wrote about the tractability of the RTRL in case of some of the RNN architectures (Quasi-RNN or Simple Recurrent Units).
- The paper does experiments for RL, where on different experiments the paper shows the method trained with RTRL can match or even sometimes outperform Truncated Backprop through time.

====

Read the author's rebuttal.

**Strengths:**

- The paper is very well written.
- The idea of using RTRL for RNNs in which element-wise recurrent is present is interesting.
- The paper does experiments on many RL tasks, and compares the proposed method to TBPTT.

**Weaknesses:**

- Since the paper is about how well RTRL works with some RNNs, it will be useful to evaluate the proposed method on more tasks like language modelling or sequential image classification etc.
- For all the experiments, it will also be useful to report a BPTT baseline i.e., where we backprop through the entire "episode".
- It will also be useful to compare to various other approximations of RTRL (using the vanilla LSTM/GRU) or even consider the tasks consider in SnAp-1.
- Not a weakness per se, but this is incorrect "Silver et al. [12] explore random projections of the sensitivity". DODGE work explores not just the random projections, but also learning the candidate direction (as a result of auxiliary tasks or via RL like synthetic gradients).

**Questions:**

- Evaluate the proposed method on more tasks like language modelling, sequential image classification as well as compare to SnAp-1.
- Report the BPTT baseline, as well as some approximation of RTRL to just see how well those approximations work for the problems chosen in this work.
- It will be helpful to make sure similar setup is followed throughout the experiments. For example: for some experiments, the paper uses the pre-trained visual encoder, or trains the visual encoder via the truncated BPTT gradient.

**Limitations:**

See Weaknesses and Questions.

---

> ### Author Rebuttal · Authors · 2023-08-09
>
> We thank the reviewer for spending valuable time on reviewing our work. We also thank the reviewer for many positive comments (soundness, presentation, and contribution scores are all “3 good”).
> We are not sure which aspects led the reviewer to rate our work as 4 with confidence 5 in the end, despite these good scores, but anyway, we believe we have good explanations to resolve all the concerns raised by the reviewer.
> If you find our response convincing, please consider increasing the score.
>
> > *"It will also be useful to compare to various other approximations of RTRL"*
>
> > *"it will be useful to evaluate the proposed method on more tasks like language modelling or sequential image classification etc."*
>
> Thank you for these valuable comments. Please find our explanations in the general response.
>
> > *For all the experiments, it will also be useful to report a BPTT baseline i.e., where we backprop through the entire "episode".*
>
> Please notice that full BPTT will yield the same performance as RTRL (using the same M), since they are *mathematically* equivalent (they compute exactly the same gradients), because our RTRL has no approximation and we use the same update frequency.  In fact, we even make use of this equivalence to test the correctness of our RTRL implementation (provided in the supplemental material; see `code/diagnostic/rtrl_layers.py`).
>
> We also would like to draw the reviewer’s attention to the fact that the requested experiment is partially/almost provided in the case of DMLab-30 Select-Non-Matching-Object (Table 1). Since the mean episode length is 100 for this task (see Line 217), by setting M=100, TBPTT is almost equal to full BPTT here. As expected by the mathematical equality, RTRL and TBPTT empirically perform similarly in this case.
> Therefore, unless the reviewer still thinks that such extra experiments are crucial (if so, please explain why), we argue that the case of DMLab-30 Select-Non-Matching-Objects already illustrates the limit case of BPTT well, and that mathematically, the requested comparison is unnecessary in general (the requested baseline will perform equally to RTRL with the same M).
>
> > *It will be helpful to make sure similar setup is followed throughout the experiments. For example: for some experiments, the paper uses the pre-trained visual encoder, or trains the visual encoder via the truncated BPTT gradient.*
>
> Training all DMLab agents from scratch was unfortunately not realistic given our compute budget (to clarify: DMLab is very CPU demanding, which was problematic for our 8-GPU nodes; running two DMLab instances was making the entire node unusably slow). We’d like to argue that this is not a crucial aspect for the main goal of this work, as we anyway cannot train the vision module using RTRL (as explained in the limitation section; Line 298) and our main focus is on the memory module.
>
> > *Not a weakness per se, but this is incorrect "Silver et al. [12] explore random projections of the sensitivity". DODGE work explores not just the random projections, but also learning the candidate direction (as a result of auxiliary tasks or via RL like synthetic gradients).*
>
> Thank you for pointing this out.
> We suppose the reviewer meant "not complete" instead of *"incorrect"* (since they do explore random projections).
> We agree with the reviewer that this sentence can be misleading.  We will replace it by "... explore projection-based approximations of the sensitivity".
>
> We hope our response brings clarifications to all of the reviewer’s remaining concerns.
> In fact, the overall rating of 4 with confidence 5 is quite surprising when soundness, presentation, and contribution scores are all “3 good”. If you find our response useful and/or convincing, please consider largely increasing the score. Thank you.

---

> > ### Comment · Reviewer_5hxy · 2023-08-17
> > **Response**
> >
> > I thank the authors for taking time and replying to the review.
> >
> > "In fact, the overall rating of 4 with confidence 5 is quite surprising when soundness, presentation, and contribution scores are all “3 good”"
> >
> > I understand the goal is to just study the RTRL in which no approximation is needed. It's hard for the reviewer to know how much important "no approximation" is unless until authors actually evaluate baselines which actually do approximate RTRL in different ways. The authors have done a comprehensive job, so the reviewer is puzzled why authors think reporting baselines is not central to the study.

---

> > > ### Comment · Reviewer_5hxy · 2023-08-17
> > > **misleading**
> > >
> > > "We agree with the reviewer that this sentence can be misleading."
> > >
> > > Yes, the reviewer agrees. The reviewer just meant that the sentence can be misleading. Thanks for correcting!

---

> > > ### Author Response · Authors · 2023-08-17
> > > **Thank you for your response**
> > >
> > > Thank you very much for your response!
> > >
> > > We would like to clarify one subtlety: we do not disagree with the reviewer that comparing "full vs approximate RTRL" would be useful. However, comparing LSTM+SnAp-1 (an example approximate RTRL suggested by the reviewer) and our eLSTM+RTRL would not allow us to directly draw any useful conclusion, because that changes both the model architecture (LSTM vs eLSTM) and the learning algorithm (SnAp-1 vs. full RTRL).
> > > For example, on Watermaze, there is a significant performance gap between LSTM vs eLSTM even when both models are trained with TBPTT (eLSTM outperforms LSTM; please refer to the extra experiment in the global response).
> > >
> > > This is why we wrote in our global response (under "To Reviewers 4QGx and 5hxy [comparison to other methods]"):
> > >
> > > > *in contrast, "full" vs. "approximate" RTRL for eLSTM could have been an interesting comparison, but we do not have such an "approximate RTRL" for eLSTM.*
> > >
> > > Since RTRL does not require any approximation in our case, our evaluation focuses on how much it improves over the TBPTT baseline.
> > > Nevertheless, if the reviewer still thinks that numbers for LSTM+SnAp-1 (or for any other methods) are crucial in our study, we'll be happy to discuss further.
> > >
> > > Regarding:
> > > > *how much important "no approximation"*
> > >
> > > we would like to emphasize that the significance of not needing to worry about approximation is not just empirical but also conceptual.
> > > Not only did it enable us to evaluate the exact RTRL (for eLSTM) unlike any prior works, but it also allowed us to look ahead and discuss RTRL beyond "approximation methods'" (see our Sec 5 on Limitations and Discussion).
> > > We critically discussed further challenges and other facets of RTRL which (to the best of our knowledge) have never been discussed in prior RTRL works focused on approximation methods.
> > > (Also, potentially, our observation that certain well-known RNNs such as Quasi-RNN can be trained using RTRL with no approximation in the one-layer case might open up new avenues in RTRL research.)
> > >
> > > We argue that all these are also our important contributions to general RTRL research (driven from the "no approximation" case).

---

> > > > ### Author Response · Authors · 2023-08-20
> > > > **SnAp-1 results**
> > > >
> > > > > Nevertheless, if the reviewer still thinks that numbers for LSTM+SnAp-1 (or for any other methods) are crucial in our study, we'll be happy to discuss further.
> > > >
> > > > While waiting for the reviewer's response, we implemented SnAp-1 for a variant of the eLSTM architecture where we replaced the element-wise recurrence in eLSTM by the full recurrence (i.e, weight vectors w^f and w^z in Eq 5 are replaced by weight matrices) to compare with our eLSTM+RTRL. The results we obtained on the DMLab environments using M=100 are as follows:
> > > >
> > > > | system | Select-Non-Matching-Object | Watermaze |
> > > > | ---------- | ------------------------|------------|
> > > > | fully-recurrent-eLSTM + **SnAp-1** | 53.6 +- 0.8  | 27.7* +- 3.2 |
> > > > | eLSTM + **full RTRL** (Table 1) | 62.2 +- 0.3 | 54.8 +- 4.3 |
> > > >
> > > > (* we found that on the challenging Watermaze, SnAp-1 variant is even unstable: its peak performance is obtained around 55M steps after which the return even goes downward, while TBPTT and full RTRL exhibit stable/monotonic score increase for 100M steps; indeed this also confirms that it is unreasonable to work with inexact gradients in challenging RL tasks).
> > > >
> > > > This effectively shows that eLSTM+full RTRL is better than fully-recurrent-eLSTM+SnAp-1
> > > > (we will improve the name of the latter model, since it is not "element-wise" anymore; for now left as-is to emphasise its connection to the eLSTM architecture).
> > > >
> > > > As we argued before, it is not possible to call this a fair/meaningful comparison between two learning algorithms (due to the different base architectures used; element-wise vs. fully recurrent eLSTM), but this is the closest architecture to our eLSTM that requires an approximation for tractable RTRL, and we agree with the reviewer that such comparison adds extra value to our work. If the reviewer confirms this is useful, we promise to run similar experiments for M=10 and M=50 cases (as in Table 1) and will add the results to the final version of the paper (and include the SnAp-1 implementation in our code which we will release publicly).
> > > >
> > > > At the same time, we'd like to stress that our main result remains full RTRL vs TBPTT. Direct comparison of two methods/algorithms without any approximation is the most basic result/question, which has been missing thus far in the (modern) RTRL literature beyond toy tasks.
> > > >
> > > > Overall, we really appreciated the reviewer's clarification/engagement. Unless you still think there are reasons for rejection that outweigh our contributions, we’d very much appreciate it if you could consider increasing your score.
> > > > Thank you very much.

---

> > > > > ### Comment · Reviewer_5hxy · 2023-08-20
> > > > > **Thanks**
> > > > >
> > > > > "At the same time, we'd like to stress that our main result remains full RTRL vs TBPTT. Direct comparison of two methods/algorithms without any approximation is the most basic result/question, which has been missing thus far in the (modern) RTRL literature beyond toy tasks"
> > > > >
> > > > > Well, it's not just full RTRL v/s TBPTT. The authors does this in the context of element-wise recurrence.
> > > > >
> > > > > Thanks for conducting extra experiments. These are helpful, but this does not address my main criticism. I was hoping that the authors with use the same setup as the baselines (i.e., SnAp-1 using LSTM/GRU as compared to element wise recurrence).
> > > > >
> > > > > The basic result which I want to see is: if the tasks are actually easier, may be LSTM with SnAP-1 approximation actually does just fine on the tasks which authors considered. As authors mentioned, no one has done this basic ablation, so it's hard for the reviewer to know. For example in procgen section authors wrote "However, in our preliminary experiments, we observe that even in these POMDP settings, both the feedforward and LSTM baselines perform similarly" so, I just want to see how LSTM and some approximation to RTRL performs.
> > > > >
> > > > > "we’d very much appreciate it if you could consider increasing your score"
> > > > >
> > > > > Will very much appreciate, if authors don't mention it in every comment..

---

> > > > > > ### Author Response · Authors · 2023-08-20
> > > > > > **Clarifying our SnAp-1 experiments**
> > > > > >
> > > > > > Thank you so much again for your reply. There seems to be some misunderstanding about the conducted experiments.
> > > > > >
> > > > > > > *"I was hoping that the authors with use the same setup as the baselines (i.e., SnAp-1 using LSTM/GRU as compared to element wise recurrence)."*
> > > > > >
> > > > > > That is what we did. The new experiment with SnAp-1 is done with a variant of LSTM, i.e., a recurrent neural network with **full recurrence** (the exact architecture is actually closer to GRU as we clarify below). Here is again its description:
> > > > > >
> > > > > > > (we wrote) *"we implemented SnAp-1 for a variant of the eLSTM architecture where we replaced the element-wise recurrence in eLSTM by the full recurrence (**i.e, weight vectors w^f and w^z in Eq 5 are replaced by weight matrices**)"*
> > > > > >
> > > > > > > (we wrote) *'we will improve the name of the latter model, since it is not "element-wise" anymore.'*
> > > > > >
> > > > > > The resulting equations are:
> > > > > >
> > > > > > $\mathbf{f}(t)  = \sigma(\mathbf{F}\mathbf{x}(t) + \mathbf{W}^{f} \mathbf{c}(t-1))$ [new Eq 5 with full recurrence]
> > > > > >
> > > > > > $\mathbf{z}(t)  = \tanh(\mathbf{Z}\mathbf{x}(t) + \mathbf{W}^{z} \mathbf{c}(t-1))$ [new Eq 5 with full recurrence]
> > > > > >
> > > > > > $\mathbf{c}(t)  = \mathbf{f}(t) \odot \mathbf{c}(t-1) + (1 - \mathbf{f}(t)) \odot \mathbf{z}(t)$ [Eq 6 unchanged]
> > > > > >
> > > > > > Followed by the feedforward components:
> > > > > >
> > > > > > $\mathbf{o}(t)  = \sigma(\mathbf{O}\mathbf{x}(t)  + \mathbf{W}^{o} \mathbf{c}(t))$ [Eq 7 unchanged]
> > > > > >
> > > > > > $\mathbf{h}(t) = \mathbf{o}(t) \odot \mathbf{c}(t) $ [Eq 7 unchanged]
> > > > > >
> > > > > > We opted for this architecture because, since the LSTM baseline underperforms the eLSTM on these tasks when both are trained using TBPTT (reported in our global response), it is "unfair" to compare LSTM using an approximate RTRL/SnAp-1 vs. eLSTM using full RTRL (**both the architecture and the learning algorithm are disadvantageous in the former system**).
> > > > > >
> > > > > > To evaluate the SnAp-1 algorithm, it makes more sense to use a fully recurrent architecture that is (at least) close to our eLSTM. **The resulting architecture above is a variant of GRU (single recurrent state vector, input-forget gate tying) which itself is a variant of LSTM.**
> > > > > >
> > > > > > > *"if the tasks are actually easier, may be LSTM with SnAP-1 approximation actually does just fine on the tasks which authors considered."*
> > > > > >
> > > > > > **Our experiments show that this is effectively not the case: "full RTRL with element-wise recurrence" outperforms "SnAp-1 with a fully recurrent neural network."**
> > > > > >
> > > > > > > *"For example in procgen section authors wrote "However, in our preliminary experiments, we observe that even in these POMDP settings, both the feedforward and LSTM baselines perform similarly" so, I just want to see how LSTM and some approximation to RTRL performs."*
> > > > > >
> > > > > > This description is irrelevant to the current discussion. **This sentence explains why we did NOT use any of the ProcGen memory-mode environments in our experiments**. In fact, six of the ProcGen environments have the so-called "memory mode", which, based on its name, seems ideal for testing recurrent policies. But as it turns out, this is not the case: feedforward and recurrent policies perform similarly on these tasks (we also had a related discussion on this in our response to Reviewer 4QGx). Therefore, our work opted for NOT using these environments in which there is no hope to show the benefit of recurrent policies anyway.
> > > > > >
> > > > > > > *"Will very much appreciate, if authors don't mention it in every comment.. "*
> > > > > >
> > > > > > We are very sorry, we stop doing it.
> > > > > > We also apologize if there was anything confusing in the original description of our SnAp-1 experiments.

---

> > > > > > > ### Author Response · Authors · 2023-08-21
> > > > > > > **Discussion period ends; Thank you for the engagement**
> > > > > > >
> > > > > > > It seems that Reviewer 5hxy did not have enough time to respond to our message above aimed at resolving his/her misunderstandings about our SnAp-1 experiments, before the end of the discussion period.
> > > > > > >
> > > > > > > We understand, and we thank the reviewer once again for his/her overall engagement.
> > > > > > >
> > > > > > > Our essential message was that: the fully recurrent architecture we utilised is similar to the vanilla LSTM/GRU (the core misunderstanding we were trying to resolve), and that ours is more pertinent to the comparison in question than any other LSTM/GRU architectures, because of (1) its close resemblance to the eLSTM architecture employed in the main experiment evaluating RTRL (making the comparison as fair as possible), and (2) the weak performance of LSTM when compared to eLSTM in these tasks, as we have already demonstrated using TBPTT.
> > > > > > >
> > > > > > > There was also one critical misunderstanding regarding the ProcGen "memory" tasks (we did NOT use them!). However, we expect this confusion will also be resolved without requiring any further discussions, since this is a simple factual misunderstanding. Our SnAp-1 experiments confirm that the DMLab environments are indeed challenging tasks where the use of inexact gradients is clearly harmful.

---

### Official Review · Reviewer_4QGx · 2023-07-06

**Soundness:** 1 poor
**Presentation:** 3 good
**Contribution:** 3 good
**Rating:** 4
**Confidence:** 4

**Summary:**

The paper modifies the LSTM architecture so that RTRL complexity, which is usually O(n^4), becomes O(n^2) where n is the number of recurrent units. This is achieved by constraining the recurrent weights to a diagonal structure. After performing some diagnostic tests, the paper presents experimental results using the proposed eLSTM in actor-critic RL settings, specifically DMLab, ProcGen, and Atari. The paper concludes with a discussion on the complexity of RTRL in the multi-layer case.


**Strengths:**

The paper is well written and clearly structured.

While the ideas presented in the paper are not novel per se, the application of these ideas to a modern RNN architecture in the context of RL is quite novel.

The work is quite relevant for the RNN/RL communities as unlocking RTRL for large-scale recurrent models is a long-standing problem and RL is an important field of application for RNNs.

The paper discusses some rarely addressed limitations about the complexity of RTRL in the multi-layer case as well as some other interesting points.


**Weaknesses:**

The main weakness of the paper is the empirical evaluation of the method. This ranges from an incrompehensible selection of tasks over a severe lack of external baselines to insufficient evaluation methods (e.g., lack of significance testing). In the following, I will elaborate in greater detail on these and other shortcomings.

On Procgen and Atari, the paper lacks comparison to state-of-the-art methods or any other external baselines. In general, the authors seem to confuse the concepts of ablation studies and external baselines. For instance, TBPTT on the eLSTM architecture is an ablation study while TBPTT on the fully recurrent LSTM architecture is a baseline (which is missing in all experiments but should be included; see, e.g., [3] for design choices). The authors should compare to state-of-the-art baselines in all environments, not just DMLab-30.

The paper should experimentally explore the limitations arising from the diagonal constraint on the hidden-to-hidden interactions. This limitation competes with the bias in the gradient estimate used by the companion learning algorithm and SnAp-1 for LSTM. The authors should compare these RTRL variants in experiments to justify their proposed method as opposed to existing ones.

The core mechanism described in Eq. (5) that reduces the RTRL complexity has been proposed in [1] and should be cited properly.

In all RL experiments, the authors report mean and standard deviation over only 3 seeds (at least for DMLab-30 and Procgen; for Atari I didn't find any info) without any significance testing. In particular, the use of the standard deviation as a measure of uncertainty in combination with a low number of seeds has been critisized as bad practice and should be replaced by interval estimates such as IQM [2].

The paper states that it focuses on "RTRL-based algorithms beyond diagnostic tasks." However, it limits these experiments to the realm of RL, where the importance of memory is often unclear. The authors should investigate how the architectural changes affect performance in some standard supervised tasks.

[1] Felix A. Gers, Jürgen Schmidhuber:
Recurrent Nets that Time and Count. IJCNN (3) 2000: 189-194

[2] Rishabh Agarwal, Max Schwarzer, Pablo Samuel Castro, Aaron C. Courville, Marc G. Bellemare:
Deep Reinforcement Learning at the Edge of the Statistical Precipice. NeurIPS 2021: 29304-29320

[3] Tianwei Ni, Benjamin Eysenbach, Ruslan Salakhutdinov:
Recurrent Model-Free RL Can Be a Strong Baseline for Many POMDPs. ICML 2022: 16691-16723


**Questions:**

The proposed eLSTM architecture does not use any cell state activation function. This does not affect the complexity of RTRL so I wonder why this slightly off-standard decision was made.

Line 21, 53, and 290: shouldn't the word "quadratic" be replaced by "quartic"?

Why did the authors dedicate resources to reactive tasks where no benefit are to be expected or an RNN might not be helpful to begin with? I had the impression that resources for experiments are limited. This exacerbates the question why reactive tasks were explored.

The paper states its focus is "to evaluate learning with untruncated gradients rather than the potential for online learning." I wonder on what basis this decision was made as online updates could lead to high sample efficiency, one of the key problems in RL. I feel the paper missed a big chance by not researching this connection of online updates and sample efficiency. Instead, it focussed on reactive tasks.

The paper states that "both the feedforward and LSTM baselines perform similarly" in Procgen Memory mode. However, [5] shows that the feedforward baseline is outperformed by methods using memory in 4/6 environments. Moreover, the Procgen paper [4] (see appendix H) shows that LSTM or framestacking outperforms feedforward in 4/6 envirnments even in non-memory mode.

In summary, I really like the presented method (and the way it was presented) and believe that it bears quite some potential. However, the empirical evaluation is very weak and I do not believe it can be fixed in the short rebuttal period. Therefore, publication of the paper seems premature at this point.

[4] Karl Cobbe, Christopher Hesse, Jacob Hilton, John Schulman:
Leveraging Procedural Generation to Benchmark Reinforcement Learning. ICML 2020: 2048-2056

[5] Fabian Paischer, Thomas Adler, Vihang P. Patil, Angela Bitto-Nemling, Markus Holzleitner, Sebastian Lehner, Hamid Eghbal-Zadeh, Sepp Hochreiter:
History Compression via Language Models in Reinforcement Learning. ICML 2022: 17156-17185

**Limitations:**

The authors give some interesting discussion on the limitation of their work, in particular the multi-layer RNN case.

---

> ### Author Rebuttal · Authors · 2023-08-09
>
> We thank the reviewer for spending valuable time on reviewing our work.
> There are several misunderstandings to be resolved, but we believe our explanations will clarify all the concerns raised by the reviewer.
>
> The main criticism breaks down into three points (quoting the reviewer):
>
> 1. > *“incrompehensible selection of tasks”*
>
> 2. > *“a severe lack of external baselines”*
>
> 3. > *“insufficient evaluation methods (e.g., lack of significance testing)”*
>
> **Regarding 1.**, we use DMLab (tasks that explicitly test memory), ProcGen/Chaser (also explicitly benefiting from memory), and subsets of Atari (following the R2D2 paper which also focuses on recurrent policies). We only added one DMLab “reactive” task to ensure that there is no unexpected downsides of RTRL in this case.
> Given that these tasks are widely used in our community (as we note in Sec 4.2 and 4.3), it seems a bit exaggerated, when the reviewer calls these choices *“incomprehensible”* (which is an unnecessarily offensive expression in our opinion), or when s/he claims that we *"focussed on reactive tasks."*
>
> **Regarding 2.**, please refer to our general response.
>
> **Regarding 3.**, we argue that the reviewer's claim *“insufficient evaluation”* is an overstatement due to an extreme interpretation of Reference [2] (reviewer’s numbering). It is not as if we only reported point estimates: we do report per-task mean scores with standard deviations. Sec 4.2 in [2] says *“Although tables containing per-task mean scores and standard deviations can reveal this variability, such tables tend to be overwhelming for more than a few tasks. In addition, standard deviations are sometimes omitted.”*  In fact, two of the main criticisms against per-task tables in [2] are: (1) overwhelming for a large number of tasks, (2) std is often omitted. In our case, both (1) and (2) are addressed (we have *“a few tasks"* and report stds). As for IQM suggested by the reviewer, [2] recommends to use IQM in the case where we aggregate scores across various tasks (see the summary in Table 1 in [2]), e.g., over 57 Atari games. This is not our case (of course, we are not saying that we are against such a metric in other scenarios with outliers).
>
> Regarding the number of seeds, we agree that more would be nicer, but we used 3 following the R2D2 paper. Unlike R2D2, we do report stds. (Regarding Atari, the reviewer wrote *“for Atari I didn't find any info”* but we comment about this in Appendix B4, Line 673).
>
> Since we have all individual numbers, we will be happy to report any additional meaningful metrics, but overall, as far as our experiments are concerned, we argue that the mean/std per-task table is not as bad as the reviewer claims, even considering [2].
>
> > *"The paper states that "both the feedforward and LSTM baselines perform similarly" in Procgen Memory mode. However, [5] shows that the feedforward baseline is outperformed by methods using memory in 4/6 environments."*
>
> We really appreciate the reviewer's effort checking this, but the reviewer's claim about [5] is wrong.
> The reviewer should have mixed up something while reading [5].
> Figure 6 of [5] also shows that, overall, the LSTM baseline (blue curve; “Impala-PPO”) performs "similarly" to the feedforward baseline (orange; “CNN-PPO” i.e., “Impala-PPO” without LSTM) (very slightly better on 2/6 tasks, Caveflyer/Maze, but similar).
>
> > *"The core mechanism described in Eq. (5) that reduces the RTRL complexity has been proposed in [1] and should be cited properly."*
>
> Thank you for pointing out this missing citation! However, the reviewer’s claim *“The core mechanism described in Eq. (5) that reduces the RTRL complexity”* is not correct. This is **not at all** the core mechanism allowing for the RTRL complexity reduction. This is an arbitrary decision (we followed Simple Recurrent Units). What allows us to reduce the complexity is the **removal** of the full recurrence. For example, Quasi-RNNs do not have such connections, but they still allow for tractable RTRL.
>
> > *"The proposed eLSTM architecture does not use any cell state activation function."*
>
> Yes, this is an arbitrary decision: we simply followed the Quasi-RNN architecture.
>
> > *""quadratic" be replaced by "quartic"?"*
>
> Yes, absolutely! Thank you!
>
> > *"The paper states its focus is "to evaluate learning with untruncated gradients rather than the potential for online learning." I wonder on what basis this decision was made"*
>
> Evaluating truncated vs. untruncated gradients is our priority because gradient truncation is the fundamental limitation of TBPTT as a gradient-based algorithm.
>
> Regarding online learning, please notice that our experiments do provide certain empirical findings. In Table 1, smaller M corresponds to more frequent updates (more “online” as M approaches 1). As expected, more online learning deteriorates the performance, likely due to sensitivity staling (compare M=10 vs. M=100). Therefore, we can expect the fully online case (M=1) to be sub-optimal in terms of final performance given a sufficient number of samples.
> Regarding the sample efficiency, at least on DMLab Select-Non-Matching-Object (Figure 1 (a) vs (d)),  we do observe sample efficiency of more “online” RTRL: at x = 2.5M steps, RTRL with M=10 achieves a score over 50, while the M=100 one is below 40. We will add a paragraph discussing these observations in the final version. Thank you for drawing our attention to this.
>
> >  *"the empirical evaluation is very weak"*
>
> We’d like to claim that, in terms of experiments, this work clearly presents a big jump compared to prior works on RTRL that report only on toy tasks. In particular, it seems that the reviewer completely ignores our main experiments on DMLab (no comment on these positive results).
>
> Overall, we believe we thoroughly responded to all the concerns raised by the reviewer. We hopefully resolved the core misunderstandings/confusions. If you find our response useful, please consider increasing the score. Thank you.

---

> > ### Comment · Reviewer_4QGx · 2023-08-15
> >
> > thank you for your response.
> >
> > I am sorry if my review seemed offensive, that was not my intention.
> >
> > I was simply surprised by the choice of experiments for the presented approach, as answering the question of “how much performance improvements can RTRL (without any approximation) bring compared to TBPTT?” on reactive tasks (and not memory-dependant tasks) seems counter-intuitive.
> >
> > Furthermore, when presenting results on a benchmark such as DMLab I would still expect a SOTA baseline, even when not comparing directly against it.
> >
> > Overall, I will maintain my score.

---

> > > ### Author Response · Authors · 2023-08-15
> > > **Thank you for your response**
> > >
> > > Thank you very much for your response!
> > >
> > > > "*I was simply surprised by the choice of experiments for the presented approach, as answering the question of “how much performance improvements can RTRL (without any approximation) bring compared to TBPTT?” on reactive tasks (and not memory-dependant tasks) seems counter-intuitive."*
> > >
> > > As we explained in our response, this was one of the reviewer's major misunderstandings. Unlike what the reviewer claimed in the original review, we do NOT focus on reactive tasks at all. We only have ONE reactive task (please refer to our first point "Regarding 1" in our response). All other tasks are chosen to test memory (and/or used by prior works testing memory).  Since this is a **factual misunderstanding**, we hope that the reviewer agrees that this issue has been resolved (we suppose this is why the reviewer wrote "I was").
> > >
> > > > *"Furthermore, when presenting results on a benchmark such as DMLab I would still expect a SOTA baseline, even when not comparing directly against it."*
> > >
> > > If the reviewer has any baseline/SOTA references on these tasks that s/he thinks we should add/cite, we'd be happy to do so. In any case, this can be easily done, and thus, this does not seem to be a major issue anymore.
> > >
> > > > *"Overall, I will maintain my score."*
> > >
> > > Based on the reviewer's response, it seems to us that our response has successfully addressed all major issues originally raised by the reviewer. So at this point, the reviewer's decision to maintain the score of "4. Borderline reject" is surprising, and the reasons for this decision are entirely unclear to us. If in the end, the reviewer concurs that s/he has no strong/valid reasons to vote for rejection anymore, please increase the score. Thank you very much.

---

> > > > ### Author Response · Authors · 2023-08-16
> > > > **Thank you for your response (continued)**
> > > >
> > > > (To add to what we wrote above:)
> > > > > If in the end, the reviewer concurs that s/he has no strong/valid reasons to vote for rejection anymore, please increase the score.
> > > >
> > > > Otherwise, we would really appreciate it a lot if the reviewer could provide compelling reasoning for her/his decision to maintain the rejection vote (4. Borderline reject), despite our thorough response resolving and correcting the reviewer's concerns and misunderstandings. The current situation does not seem fair. If there is anything in our response that the reviewer still disagrees with, we'd be happy to provide further clarifications, so please let us know. Thank you very much.

---

### Author Rebuttal · Authors · 2023-08-09

We thank all the reviewers for their valuable comments.
Overall, we believe that our response should resolve all main concerns raised by the reviewers.
If you find our explanations convincing, please consider increasing your scores.

For the moment, we will keep the originally submitted PDF as is, such that we can refer to the original line numbers. We will apply all promised edits if the paper is accepted.

While we reply to each reviewer individually below, please find some common responses here.

**To all the Reviewers, especially 4QGx and Y8rh** [vanilla LSTM results]

We conducted some extra experiments with the basic LSTM using either 1 or 2 layers (requested by Reviewer Y8rh), trained with TBPTT, on two of our main tasks, DMLab Select-Non-Matching-Object and Watermaze (comparable with Table 1) using M=100.
The results are as follows (mean +- std computed using 3 training seeds):

| Model | Select-Non-Matching-Object | Watermaze |
|----------|---------------------------------------|-----------------|
| LSTM 1-layer, TBPTT |  61.7 +- 0.5  |  37.6 +- 5.3 |
| LSTM 2-layer, TBPTT |  61.3 +- 0.3  |  39.7 +- 4.6 |
| eLSTM 1-layer, TBPTT (Table 1) | 61.7 +- 0.1 | 45.6 +- 4.7 |

We observe that the LSTM (TBPTT) performance is comparable to eLSTM (TBPTT)’s on Select-Non-Matching-Object, while eLSTM outperforms LSTM on Watermaze (empirically, eLSTM turns out to be more sample efficient here). Overall, this shows that our restriction to eLSTM has no particular downsides while working with these tasks.

**To Reviewers 4QGx and 5hxy** [comparison to other methods]

> (Reviewer 4QGx) *“the authors seem to confuse the concepts of ablation studies and external baselines”*, *“The authors should compare to state-of-the-art baselines in all environments, not just DMLab-30”*

> (Reviewer 5hxy) *"It will also be useful to compare to various other approximations of RTRL"*

We first would like to clarify that our goal is **not at all** to promote eLSTM+RTRL as an alternative to existing RTRL methods or to present a state-of-the-art system (In particular, it seems that Reviewer 4QGx read our work as a system/architecture paper. This is not the case).
Our goal is to explore/discuss the potentials and limits of RTRL (see the title) in the **ideal but restricted case** of 1-layer element-wise RNNs where we have the possibility to study RTRL without any approximation.
Different from any prior work, this setting allows us to answer the natural question:  “how much performance improvements can RTRL (without any approximation) bring compared to TBPTT?” Driven by this question, all our experiments mainly focus on the comparison between eLSTM+RTRL vs. eLSTM+TBPTT.

Strictly speaking, whether eLSTM+RTRL is better than LSTM+SnAp-1 (or any other method) is secondary to our main question above. Such a comparison would only tell us which “system” is better (in contrast, "full" vs. "approximate" RTRL for eLSTM could have been an interesting comparison, but we do not have such an "approximate RTRL" for eLSTM).
Similarly, comparing our Atari scores (trained only on 200M frames) with some state-of-the-art systems would not allow us to answer any scientific questions of our interest.
The very good performance we report on DMLab compared to the IMPALA and R2D2 baselines, is a **by-product** of this work, not the original objective.

Therefore, while we agree with the reviewers that the proposed comparisons would provide some extra results, we argue that they are not central to our study.
Instead, we discuss more fundamental aspects of RTRL research in general, looking ahead to further challenges beyond the current research on RTRL focused on comparing approximation methods (Reviewer Bqpj wrote this is *“interesting and forward-looking”*).
We really hope that this clarifies our scope and contributions.

**To Reviewers 4QGx and 5hxy** [supervised learning experiments]

> (Reviewer 4QGx) *"The paper states that it focuses on "RTRL-based algorithms beyond diagnostic tasks." However, it limits these experiments to the realm of RL"*

> (Reviewer 5hxy) *"... more tasks like language modelling or sequential image classification"*

First of all, limiting experiments to RL tasks is not at all in contradiction with our claim of going *“beyond diagnostic tasks.”* For example, DMLab is a challenging non-toy task for evaluating memory in the RL domain.
We'd like to stress that RL in POMDPs is a very important topic in our community, which is underexplored in prior work on RTRL (perhaps due to the challenge of ensuring good RTRL approximation quality while dealing with RL's practical difficulties; in contrast, we do not have to worry about the former).

Regarding supervised learning, unfortunately, our 1-layer architecture is too limited to perform well on modern language modelling tasks (dominated by deep models/Transformers) or any other *meaningful* supervised tasks at scale.
If we restrict ourselves to 1-layer models, the task will be reduced to a toy setting similar to those used in prior works, contradicting our original objective. We have an explicit paragraph discussing the limitation of this architecture in Appendix B.1./Line 628, and some comments on language modelling in Line 638 (supported by Reviewer Bqpj: *“It is nice to see the authors be honest about the limitations of RTRL throughout the paper.”*), but we'll add an explicit discussion on this aspect in the final version.

As a general note: for RTRL to scale beyond toy tasks in supervised learning, it would require an optimised implementation such as a custom CUDA kernel (to be competitive with today's highly optimised TBPTT implementations).
This would require further engineering efforts (but also some thoughts on the algorithm too: in Line 324 we mention the RTRL-BPTT hybrid as a more promising RTRL variant for supervised learning), which we leave for future work.

In sum, these observations led us to focus on the RL tasks where one-layer recurrent policy is common.

---

> ### Author Response · Authors · 2023-08-21
> **​Summary of the additional experimental results**
>
> Given that these results might also interest other reviewers, here we provide a recap of the new experimental results we presented in our discussion with Reviewer 5hxy.
>
> **Description**
>
> We conducted extra experiments using SnAp-1 (an approximate RTRL algorithm) on a fully recurrent neural network (obtained by replacing element-wise recurrence in the eLSTM by the full recurrence; going back to an architecture similar to the standard LSTM/GRU) on the DMLab environments with M=100.
>
> **Results**
>
> | RNN Architecture | Learning algorithm | Select-Non-Matching-Object | Watermaze |
> | ---------- | ------------------------|------------|------------|
> | Fully recurrent | SnAp-1 | 53.6 +- 0.8  | 27.7 +- 3.2 |
> | Element-wise recurrent |  RTRL (Table 1) | 62.2 +- 0.3 | 54.8 +- 4.3 |
>
> - This effectively shows that "full RTRL with element-wise recurrence" outperforms "SnAp-1 with a fully recurrent neural network."
>
> - On the challenging Watermaze task, we even found that SnAp-1 is unstable: its peak performance is obtained around 55M steps after which the return even goes downward, while TBPTT and full RTRL exhibit stable/monotonic score increase for 100M steps. Indeed, this also confirms that it is unreasonable to work with inexact gradients in challenging RL tasks.
>
> **Remarks**
>
> (As we have already noted in our previous responses) Strictly speaking, this is not a "fair" comparison between approximate vs. full RTRL algorithms.
> The base RNN architectures used in the two systems are different (full vs. element-wise recurrence) and we know that the choice of the base RNN architecture itself influences the performance: we have already shown that eLSTM outperforms LSTM on these tasks.
> To minimize this architectural discrepancy, our fully recurrent architecture is obtained by *minimally* modifying the eLSTM architecture (just by replacing its element-wise recurrent components by its fully recurrent counterparts).

---

### Decision · Program_Chairs · 2023-09-21

**Decision:**

Reject

**Comment:**

### Summary

This paper introduces a variation of LSTM architecture called eLSTM, which reduces the real-time recurrent Learning (RTRL) complexity from $O(n^4)$ to $O(n^2)$, where $n$ represents the number of recurrent units. This speedup is achieved by imposing a diagonal structure on the recurrent weights. The authors discuss the tractability of RTRL in various RNN architectures, including Quasi-RNN and Simple Recurrent Units. The paper conducts experiments in actor-critic reinforcement learning (RL) settings using eLSTM on DMLab, ProcGen, and Atari benchmarks. The results indicate that the method trained with RTRL can sometimes match or even outperform Truncated Backprop through time (TBPTT). The use of RTRL is particularly advantageous when TBPTT is short. Although the paper's experiments do not strictly necessitate RTRL, it suggests potential applications in scenarios with extensive time horizons, such as POMDPs spanning many timesteps.

### Decision

The paper is interesting and making RTRL efficient + performing well would enable lots of interesting applications of RNNs in real-world application. The proposed algorithm is interesting and technically correct. However, during the discussions after the rebuttal, the reviewers 5hxy and 4QGx insisted that this paper is not ready for publication. The main issues of this paper is in the experiments:

1. The experiments mainly focus on the comparison between eLSTM+RTRL vs. eLSTM+TBPTT", i.e., they do not really compare to external baselines. As it stands now, the baselines in this paper are quite limited.
2. The paper "focuses on reactive tasks."
3. Lack of enough seeds to do significance testing of the results.

I would recommend the authors to fix the issues pointed out by the reviews and resubmit to another venue.